# DEEP EVIDENTIAL REGRESSION

## ABSTRACT

Deterministic neural networks (NNs) are increasingly being deployed in safety critical domains, where calibrated, robust and efficient measures of uncertainty are crucial. While it is possible to train regression networks to output the parameters of a probability distribution by maximizing a Gaussian likelihood function, the resulting model remains oblivious to the underlying confidence of its predictions. In this paper, we propose a novel method for training deterministic NNs to not only estimate the desired target but also the associated evidence in support of that target. We accomplish this by placing evidential priors over our original Gaussian likelihood function and training our NN to infer the hyperparameters of our evidential distribution. We impose priors during training such that the model is penalized when its predicted evidence is not aligned with the correct output. Thus the model estimates not only the probabilistic mean and variance of our target but also the underlying uncertainty associated with each of those parameters. We observe that our evidential regression method learns well-calibrated measures of uncertainty on various benchmarks, scales to complex computer vision tasks, and is robust to adversarial input perturbations.

## 1 INTRODUCTION

Recent advances in deep supervised learning have yielded super human level performance and precision. While these models empirically generalize well when placed into new test enviornments, they are often easily fooled by adversarial perturbations (Goodfellow et al., 2014), and have difficulty understanding when their predictions should not be trusted. Today, regression based neural networks (NNs) are being deployed in safety critical domains of computer vision (Godard et al., 2017) as well as in robotics and control (Bojarski et al., 2016) where the ability to infer model uncertainty is crucial for eventual wide-scale adoption. Furthermore, precise uncertainty estimates are useful both for human interpretation of confidence and anomaly detection, and also for propagating these estimates to other autonomous components of a larger, connected system.

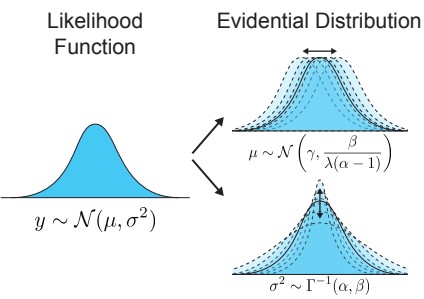

Figure 1: **Evidential distributions.** Maximum likelihood optimization learns a likelihood distribution given data, while evidential distributions model higher-order probability distribution over the likelihood parameters.

Existing approaches to uncertainty estimation are roughly split into two categories: (1) learning aleatoric uncertainty (uncertainty in the data) and (2) epistemic uncertainty (uncertainty in the prediction). While representations for aleatoric uncertainty can be learned directly from data, approaches for estimating epistemic uncertainty focus on placing probabilistic priors over the weights and sampling to obtain a measure of variance. In practice, many challenges arise with this approach, such as the computational expense of sampling during inference, how to pick an appropriate weight prior, or even how to learn such a representation given your prior.

Instead, we formulate learning as an evidence acquisition process, where the model can acquire evidence during training in support of its prediction (Sensoy et al., 2018; Malinin & Gales, 2018). Every training example adds support to a learned higher-order, *evidential* distribution. Sampling from this distribution yields instances of lower-order, likelihood functions from which the data was drawn (cf. Fig. 1). We demonstrate that, by placing priors over our likelihood function, we can learn a grounded representation of epistemic and aleatoric uncertainty without sampling during inference.

In summary, this work makes the following contributions:

1. A novel and scalable method for learning representations of epistemic and aleatoric uncertainty, specifically on regression problems, by placing evidential priors over the likelihood;

2. Formulation of a novel evidential regularizer for continuous regression problems, which we show is necessary for expressing lack of a evidence on out-of-distribution examples;

3. Evaluation of learned epistemic uncertainty on benchmark regression tasks and comparison against other state-of-the-art uncertainty estimation techniques for neural networks; and

4. Robustness evaluation against out of distribution and adversarially perturbed test data.

## 2 MODELLING UNCERTAINTIES FROM DATA

### 2.1 PRELIMINARIES

Consider the following supervised optimization problem: given a dataset, $\mathcal{D}$, of $N$ paired training examples, $(x_1, y_1), \ldots, (x_N, y_N)$, we aim to learn a function $f$, parameterized by a set of weights, $\boldsymbol{w}$, which approximately solves the following optimization problem:

$$\min_{\boldsymbol{w}} J(\boldsymbol{w}); \quad J(\boldsymbol{w}) = \frac{1}{N} \sum_{i=1}^{N} \mathcal{L}_i(\boldsymbol{w}), \tag{1}$$

where $\mathcal{L}_i(\cdot)$ describes a loss function. In this work, we consider deterministic regression problems, which commonly optimize the sum of squared errors, $\mathcal{L}_i(\boldsymbol{w}) = \frac{1}{2} \|y_i - f(x_i; \boldsymbol{w})\|^2$. In doing so, the model is encouraged to learn the average correct answer for a given input, but does not explicitly model any underlying noise or uncertainty in the data when making its estimation.

### 2.2 MAXIMUM LIKELIHOOD ESTIMATION

We can also approach our optimization problem from a maximum likelihood perspective, where we learn model parameters that maximize the likelihood of observing a particular set of training data. In the context of deterministic regression, we assume our targets, $y_i$, were drawn i.i.d. from a Gaussian distribution with mean and variance parameters $\theta = (\mu, \sigma^2)$. In maximum likelihood estimation, we aim to learn a model to infer $\theta = (\mu, \sigma^2)$ that maximize the likelihood of observing our targets, $y$, given by $p(y_i|\theta)$. In practice, we minimize the negative log likelihood by setting:

$$\mathcal{L}_i(\boldsymbol{w}) = -\log p(y_i | \underbrace{\mu, \sigma^2}_{\theta}) = \frac{1}{2} \log(2\pi\sigma^2) + \frac{(y_i - \mu)^2}{2\sigma^2}. \tag{2}$$

In learning the parameters $\theta$, this likelihood function allows us to successfully model the uncertainty of our data, also known as the aleatoric uncertainty. However, our model remains oblivious to the predictive model or epistemic uncertainty (Kendall & Gal, 2017).

In this paper, we present a novel approach for estimating the evidence in support of network predictions by directly learning both the inferred aleatoric uncertainty as well as the underlying epistemic uncertainty over its predictions. We achieve this by placing higher-order prior distributions over the learned parameters governing the distribution from which our observations are drawn.

## 3 EVIDENTIAL UNCERTAINTY FOR REGRESSION

### 3.1 PROBLEM SETUP

We consider the problem where our observed targets, $y_i$, are drawn i.i.d. from a Gaussian distribution now with *unknown mean and variance* $(\mu, \sigma^2)$, which we seek to probabilistically estimate. We model this by placing a conjugate prior distribution on $(\mu, \sigma^2)$. If we assume our observations are drawn from a Gaussian, this leads to placing a Gaussian prior on our unknown mean and an Inverse-Gamma prior on our unknown variance:

$$(y_1, \ldots, y_N) \sim \mathcal{N}(\mu, \sigma^2)$$
$$\mu \sim \mathcal{N}(\gamma, \sigma^2 \lambda^{-1}) \qquad \sigma^2 \sim \Gamma^{-1}(\alpha, \beta).$$

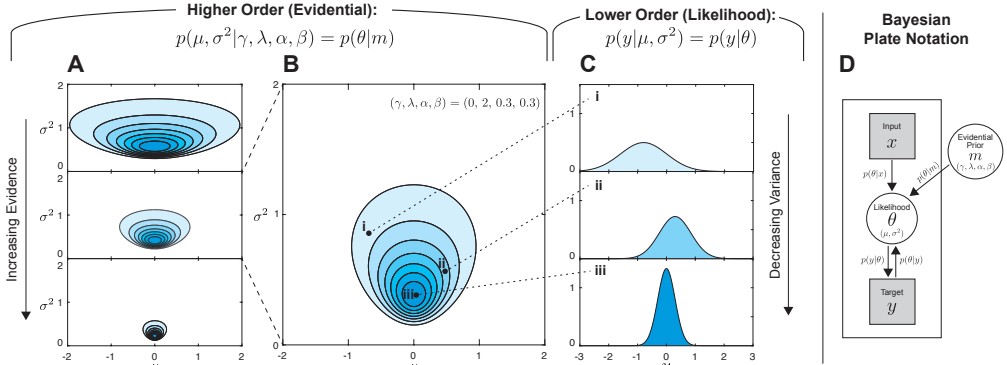

Figure 2: **Normal Inverse-Gamma distribution.** Different realizations of our evidential distribution (A) correspond to different levels of confidences in the parameters (e.g. $\mu, \sigma^2$). Sampling from a single realization of a higher-order evidential distribution (B), yields lower-order likelihoods (C) over the data (e.g. $p(y|\mu, \sigma^2)$). Darker shading indicates higher probability mass. We aim to learn a model (D) that predicts the target, $y$, from an input, $x$, with an evidential prior imposed on our likelihood to enable uncertainty estimation.

where $\Gamma(\cdot)$ is the gamma function, $m = (\gamma, \lambda, \alpha, \beta)$, and $\gamma \in R, \lambda > 0, \alpha > 0, \beta > 0$.

Our aim is to estimate a posterior distribution $q(\mu, \sigma^2) = p(\mu, \sigma^2|y_1, \dots, y_N)$. To obtain an approximation for the true posterior, we assume that the estimated distribution can be factorized (Parisi, 1988) such that $q(\mu, \sigma^2) = q(\mu)\,q(\sigma^2)$. Thus, our approximation takes the form of the Gaussian conjugate prior, the Normal Inverse-Gamma (N.I.G.) distribution:

$$p(\underbrace{\mu, \sigma^2}_{\theta} \,|\, \underbrace{\gamma, \lambda, \alpha, \beta}_{m}) = \frac{\beta^\alpha \sqrt{\lambda}}{\Gamma(\alpha)\sqrt{2\pi\sigma^2}} \left(\frac{1}{\sigma^2}\right)^{\alpha+1} \exp\left\{-\frac{2\beta + \lambda(\gamma - \mu)^2}{2\sigma^2}\right\}. \tag{3}$$

A popular interpretation of the parameters of the conjugate prior distribution is in terms of "virtual-observations" in support of a given property (Jordan, 2009). For example, the mean of a N.I.G. distribution can be interpreted as being estimated from $\lambda$ virtual-observations with sample mean $\gamma$ while its variance was estimated from $2\alpha$ virtual-observations with sample mean $\gamma$ and sum of squared deviations $2\beta$. Following from this interpretation, we define the total evidence, $\Phi$, of our evidential distributions as the sum of all inferred virtual-observations counts: ($\Phi = \lambda + 2\alpha$).

Drawing a sample $\theta_j$ from the N.I.G. distribution yields a single instance of our likelihood function, namely $\mathcal{N}(\mu_j, \sigma_j^2)$. Thus, the N.I.G. hyperparameters, $(\gamma, \lambda, \alpha, \beta)$, determine not only the location but also the dispersion concentrations, or uncertainty, associated with our inferred likelihood function. Therefore, we can interpret the N.I.G. distribution as *higher-order*, *evidential*, distribution on top of the unknown *lower-order* likelihood distribution from which observations are drawn.

For example, in Fig. 2A we visualize different evidential N.I.G. distributions with varying model parameters. We illustrate that by increasing the evidential parameters (i.e. $\lambda, \alpha$) of this distribution, the p.d.f. becomes tightly concentrated about its inferred likelihood function. Considering a single parameter realization of this higher-order distribution, cf. Fig. 2B, we can subsequently sample many lower-order realizations of our likelihood function, as shown in Fig. 2C.

In this work, we use neural networks to infer the hyperparameters of this higher-order, evidential distribution, given an input. This approach presents several distinct advantages compared to prior work. First, our method enables simultaneous learning of the desired regression task, along with aleatoric and epistemic uncertainty estimation, built in, by enforcing evidential priors. Second, since the evidential prior is a higher-order N.I.G. distribution, the maximum likelihood Gaussian can be computed analytically from the expected values of the $(\mu, \sigma^2)$ parameters, without the need for sampling. Third, we can effectively estimate the epistemic or model uncertainty associated with the network's prediction by simply evaluating the variance of our inferred evidential distribution.

### 3.2 LEARNING THE EVIDENTIAL DISTRIBUTION

Having formalized the use of an evidential distribution to capture both aleatoric and epistemic uncertainty, we next describe our approach for learning a model (c.f. Fig. 2D) to output the hyperparameters

of this distribution. For clarity, we will structure the learning objective into two distinct parts: (1) acquiring or maximizing model evidence in support of our observations and (2) minimizing evidence or inflating uncertainty when the prediction is wrong. At a high level, we can think of (1) as a way of fitting our data to the evidential model while (2) enforces a prior to inflate our uncertainty estimates.

**(1) Maximizing the model fit.**

From Bayesian probability theory, the "model evidence", or marginal likelihood, is defined as the likelihood of an observation, $y_i$, given the evidential distribution parameters $m$ and is computed by marginalizing over the likelihood parameters $\theta$:

$$p(y_i|m) = \frac{p(y_i|\theta, m)p(\theta|m)}{p(\theta|y_i, m)} = \int_\theta p(y_i|\theta, m)p(\theta|m) \, d\theta. \tag{4}$$

The model evidence is not, in general, straightforward to evaluate since computing it involves integrating out the dependence on latent model parameters:

$$p(y_i|m) = \int_{\sigma^2=0}^{\infty} \int_{\mu=-\infty}^{\infty} p(y_i|\mu, \sigma^2)p(\mu, \sigma^2|m) \, d\mu \, d\sigma^2 \tag{5}$$

However, by placing a N.I.G. evidential prior on our Gaussian likelihood function an analytical solution for the model evidence does exist. For computational reasons, we minimize the negative logarithm of the model evidence ($\mathcal{L}_i^{\text{NLL}}(\boldsymbol{w})$). For a complete derivation please refer to Sec. 7.1,

$$\mathcal{L}_i^{\text{NLL}}(\boldsymbol{w}) = -\log p(y_i|m) = -\log\left(2^{\frac{1}{2}+\alpha}\beta^\alpha \sqrt{\frac{\lambda}{2\pi(1+\lambda)}}\left(2\beta + \frac{\lambda(\gamma - y_i)^2}{1+\lambda}\right)^{-\frac{1}{2}-\alpha}\right). \tag{6}$$

Instead of modeling this loss using empirical Bayes, where the objective is to *maximize* model evidence, we alternatively can *minimize* the sum-of-squared (SOS) errors, between the evidential prior and the data that would be sampled from the associated likelihood. Thus, we define $\mathcal{L}_i^{\text{SOS}}(\boldsymbol{w})$ as

$$\mathcal{L}_i^{\text{SOS}}(\boldsymbol{w}) = \mathbb{E}_{\theta' \sim p(\theta|m)}\left[\mathbb{E}_{y' \sim p(y|\theta')}\left[||y' - y_i||_2^2\right]\right] \tag{7}$$

$$= \int_{\sigma^2=0}^{\infty} \int_{\mu=-\infty}^{\infty} \mathbb{E}_{y' \sim p(y|\mu, \sigma^2)}\left[||y' - y_i||_2^2\right] p(\mu, \sigma^2|m) \, d\mu \, d\sigma^2 \tag{8}$$

$$= \left(\frac{\Gamma(\alpha - \frac{1}{2})}{4\,\Gamma(\alpha)\,\lambda\sqrt{\beta}}\right)\left(2\beta(1 + \lambda) + (2\alpha - 1)\lambda(y_i - \gamma)^2\right). \tag{9}$$

A step-by-step derivation is given in Sec. 7.1. In our experiments, using $\mathcal{L}_i^{\text{SOS}}(\boldsymbol{w})$ resulted in greater training stability and increased performance, compared to the $\mathcal{L}_i^{\text{NLL}}(\boldsymbol{w})$ loss. Therefore, $\mathcal{L}_i^{\text{SOS}}(\boldsymbol{w})$ is used in all presented results.

**(2) Minimizing evidence on errors.**

In the first term of our objective above, we outlined a loss function for training a NN to output parameters of a N.I.G. distribution to fit our observations, either by maximizing the model evidence or minimizing the sum-of-squared errors. Now, we describe how to regularize training by applying a lack of evidence prior (i.e., maximum uncertainty). Therefore, during training we aim to minimize our evidence (or maximize our uncertainty) everywhere except where we have training data.

This can be done by minimizing the KL-divergence between the inferred posterior, q($\theta$), and a prior, $p(\theta)$. This has been demonstrated with success in the categorical setting where the uncertainty prior can be set to a uniform Dirichlet (Malinin & Gales, 2018; Sensoy et al., 2018). In the regression setting, the KL-divergence between our posterior and a N.I.G. zero evidence prior (i.e., $\{\alpha, \lambda\} = 0$) is not well defined (Soch & Allefeld, 2016), please refer to Sec. 7.2 for a derivation. Furthermore, this prior needs to be enforced specifically where there is no support from the data. Past works in classification accomplish this by using the ground truth likelihood classification (i.e., the one-hot encoded labels) to remove the non-misleading evidence. However, in regression, labels are provided as point targets (not ground truth Gaussian likelihoods). Unlike classification, it is not possible to penalize evidence everywhere except our single point estimate, as this space is infinite and unbounded. Thus, these previously explored approaches for evidential optimization are not directly applicable.

To address both of these shortcomings of past works, now in the regression setting, we formulate a novel evidence regularizer, $\mathcal{L}_i^{\text{R}}$, based on the error of the $i$-th prediction,

$$\mathcal{L}_i^{\text{R}}(\boldsymbol{w}) = \|y_i - \mathbb{E}[\mu_i]\|_p \cdot \Phi = \|y_i - \gamma\|_p \cdot (2\alpha + \lambda), \tag{10}$$

where $\|x\|_p$ represents the L-p norm of $x$. The value of $p$ impacts the penalty imposed on the evidence when a wrong prediction is made. For example, $p = 2$, heavily over-penalizes the evidence on larger errors, whereas $p = 1$ and $p = 0.5$ saturate the evidence penalty for larger errors. We found that $p = 1$ provided the optimal stability during training and use this value in all presented results.

This regularization loss imposes a penalty whenever there is an error in the prediction that scales with the total evidence of our inferred posterior. Conversely, large amounts of predicted evidence will not be penalized as long as the prediction is close to the target observation. We provide an ablation analysis to quantitatively demonstrate the added value of this evidential regularizer in Sec 7.3.2.

The combined loss function employed during training consists of the two loss terms for maximizing model evidence and regularizing evidence,

$$\mathcal{L}_i(\boldsymbol{w}) = \mathcal{L}_i^{\text{SOS}}(\boldsymbol{w}) + \mathcal{L}_i^{\text{R}}(\boldsymbol{w}). \tag{11}$$

### 3.3 Evaluating aleatoric and epistemic uncertainty

The aleatoric uncertainty, also referred to as statistical or data uncertainty, is representative of unknowns that differ each time we run the same experiment. We evaluate the aleatoric uncertainty from $\mathbb{E}[\sigma^2] = \frac{\beta}{\alpha-1}$. The epistemic, also known as the model uncertainty, describes the estimated uncertainty in the learned model and is defined as $\text{Var}[\mu] = \frac{\beta}{(\alpha-1)\lambda}$. Note that $\text{Var}[\mu] = \mathbb{E}[\sigma^2]/\lambda$, which is expected as $\lambda$ is one of our two evidential virtual-observation counts.

## 4 Experiments

### 4.1 Predictive accuracy and uncertainty benchmarking

We first qualitatively compare the performance of our approach against a set of benchmarks on a one-dimensional toy regression dataset (Fig. 3). For training and dataset details please refer to Sec. 7.3.1. We compare deterministic regression, as well as techniques using empirical variance of the networks' predictions such as MC-dropout, model-ensembles, and Bayes-by-Backprop which underestimate the uncertainty outside the training distribution. In contrast, evidential regression estimates uncertainty appropriately and grows the uncertainty estimate with increasing distance from the training data.

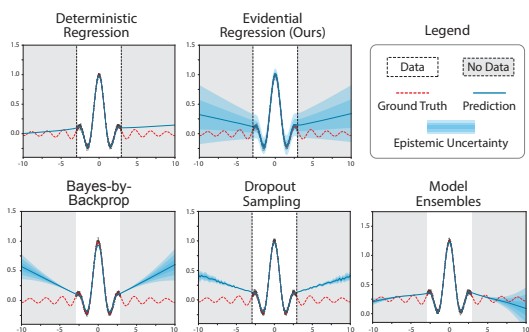

Figure 3: **Epistemic uncertainty estimation.** Modeling the supportive evidence during learning enables precise prediction within the training regime and conservative uncertainty estimates where there was no training data. Comparisons to other epistemic uncertainty estimation methods are illustrated (bottom).

Additionally, we compare our approach to state-of-the-art methods for predictive uncertainty estimation using NNs on common real world datasets used in (Hernández-Lobato & Adams, 2015; Lakshminarayanan et al., 2017; Gal & Ghahramani, 2016). We evaluate our proposed evidential regression method against model-ensembles and BBB based on root mean squared error (RMSE), and negative log-likelihood (NLL). We do not provide results for MC-dropout since it consistently performed inferior to the other baselines. The results in Table 1 indicate that although the loss function for evidential regression is more complex than competing approaches, it is the top performer in RMSE and NLL in 8 out of 9 datasets.

Furthermore, we demonstrate that, on a synthetic dataset with *a priori* known noise, evidential models can additionally estimate and recover the underlying aleatoric uncertainty. For more information please refer to Sec. 7.3.3 for results and experiment details.

| Dataset | RMSE | | | NLL | | |
|---------|------|------|-----------|------|------|-----------|
| | Ensembles | BBB | Evidential | Ensembles | BBB | Evidential |
| Boston | 0.09 ± 4.3e-4 | 0.09 ± 3.7e-4 | **0.09 ± 1.0e-6** | **-0.89 ± 6.5e-2** | -0.67 ± 1.5e-2 | **-0.87 ± 2.2e-2** |
| Concrete | **0.07 ± 4.4e-3** | **0.06 ± 3.3e-6** | **0.06 ± 7.0e-7** | **-1.29 ± 4.1e-2** | **-1.32 ± 4.3e-3** | **-1.31 ± 1.9e-2** |
| Energy | **0.10 ± 2.3e-4** | **0.10 ± 1.6e-5** | **0.10 ± 9.0e-7** | -0.61 ± 8.9e-2 | -0.60 ± 2.0e-2 | **-0.75 ± 1.4e-2** |
| Kin8nm | **0.07 ± 3.5e-4** | 0.17 ± 3.5e-4 | 0.08 ± 3.8e-3 | -0.78 ± 1.4e-2 | -0.32 ± 6.3e-3 | **-1.17 ± 2.6e-2** |
| Naval | **0.01 ± 1.0e-7** | 0.04 ± 1.2e-2 | **0.01 ± 3.4e-4** | -2.55 ± 3.3e-2 | -1.83 ± 2.4e-1 | **-3.17 ± 2.1e-3** |
| Power | **0.06 ± 4.0e-7** | **0.06 ± 2.3e-6** | **0.06 ± 5.3e-6** | -1.29 ± 6.9e-2 | -1.33 ± 2.5e-3 | **-1.40 ± 6.2e-3** |
| Protein | **0.17 ± 1.0e-6** | 0.17 ± 8.0e-4 | **0.17 ± 1.6e-6** | **-0.27 ± 6.7e-2** | 0.32 ± 5.9e-2 | **-0.29 ± 1.1e-2** |
| Wine | **0.10 ± 3.0e-4** | **0.10 ± 2.9e-4** | **0.10 ± 3.8e-5** | -0.46 ± 2.5e-1 | **-0.89 ± 2.4e-3** | -0.85 ± 6.9e-3 |
| Yacht | 0.07 ± 1.3e-3 | 0.07 ± 3.4e-3 | **0.06 ± 6.2e-5** | -1.16 ± 6.3e-2 | -0.74 ± 5.8e-2 | **-1.28 ± 9.4e-3** |

Table 1: **Benchmark regression tests.** We evaluate RMSE and negative log-likelihood (NLL) for model ensembling (Lakshminarayanan et al., 2017), Bayes-By-Backprop (BBB) (Blundell et al., 2015) and evidential regression. Evidential achieves top scores (bolded, within statistical significance) on 8 of the 9 datasets.

## 4.2 DEPTH ESTIMATION

After establishing benchmark comparison results, in this subsection we demonstrate the scalability of our evidential learning by extending to the complex, high-dimensional task of depth estimation. Monocular end-to-end depth estimation is a central problem in computer vision which aims to learn a representation of depth directly from an RGB image of the scene. This is a challenging learning task since the output target $y$ is very high-dimensional. For every pixel in the image, we regress over the desired depth and simultaneously estimate the uncertainty associated to that individual pixel.

Our training data consists of over 27k RGB-to-depth pairs of indoor scenes (e.g. kitchen, bedroom, etc.) from the NYU Depth v2 dataset (Nathan Silberman & Fergus, 2012). We train a U-Net style NN (Ronneberger et al., 2015) for inference. The final layer of our model outputs a single $H \times W$ activation map in the case of deterministic regression, dropout, ensembling and BBB. Evidential models output four final activation maps, corresponding to $(\gamma, \lambda, \alpha, \beta)$.

Table 2 summarizes the size and speed of all models. Evidential models contain significantly fewer trainable parameters than ensembles (where the number of parameters scales linearly with the size of the ensemble). BBB maintains a trainable mean and variance for every weight in the network, so its size is roughly $2\times$ larger as well. Since evidential regression models do not require sampling in order to estimate their uncertainty, their forward-pass inference times are also significantly more efficient. Finally, we demonstrate comparable predictive accuracy (through RMSE and NLL) to the other models. For a more detailed breakdown of how the number of samples effects the baselines please refer to Tab. 3. Note that the output size of the depth estimation problem presented significant learning challenges for the BBB baseline, and it was unable to converge during training. As a result, for the remainder of this analysis we compare against only spatial dropout and ensembles.

We evaluate these models in terms of their accuracy and their predictive uncertainty on unseen test data. Fig. 4A-C visualizes the predicted depth, absolute error from ground truth, and predictive uncertainty across three randomly picked test images. Ideally, a strong predictive uncertainty would capture any errors in the prediction (i.e., roughly correspond to where the model is making errors). Compared to dropout and ensembling, evidential uncertainty modeling captures the depth errors while providing clear and localized predictions of confidence. In general, dropout drastically underestimates the amount of uncertainty present, while ensembling occasionally overestimates the uncertainty.

To evaluate uncertainty calibration to the ground-truth errors, we fit receiver operating characteristic (ROC) curves to normalized estimates of error and uncertainty. Thus, we test the network's ability to detect how likely it is to make an error at a given pixel using its predictive uncertainty.

| | # Parameters | | Inference Speed | | RMSE | NLL |
|---|---|---|---|---|---|---|
| | Absolute | Relative | Seconds | Relative | | |
| **Evidential (Ours)** | 7,846,776 | 1.00 | 0.003 | 1.00 | 0.024 ± 0.032 | -1.128 ± 0.290 |
| **Spatial Dropout** | 7,846,657 | 1.00 | 0.031 | 11.48 | 0.031 ± 0.033 | -1.227 ± 0.374 |
| **Ensembles** | 39,233,285 | 5.00 | 0.010 | 3.72 | 0.023 ± 0.027 | -1.077 ± 0.298 |
| **BBB** | 11,772,929 | 1.50 | 0.070 | 25.55 | - | - |

Table 2: **Benchmark performance comparison on depth prediction.** For fair performance comparison, sampling methods were all parallelized and sampled 5 times as RMSE and NLL did not significantly improve with greater samples. For an extended analysis with larger number of samples please refer to Table 3.

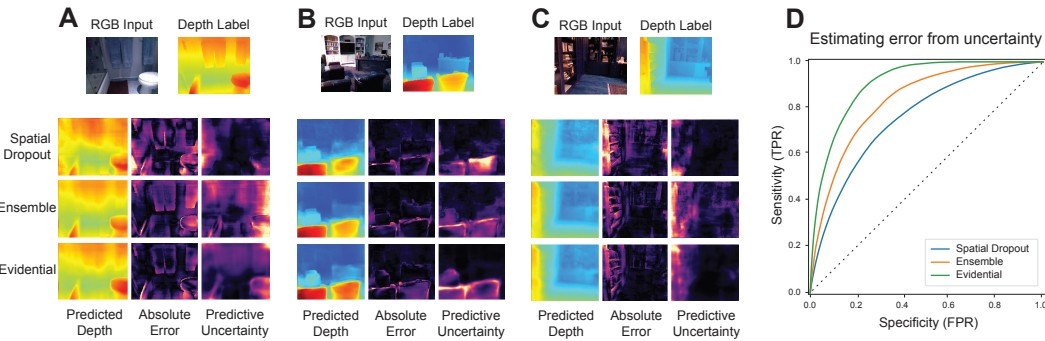

Figure 4: **Modeling uncertainty in depth estimation.** Three methods for estimating epistemic (model) uncertainty are evaluated in the context of monocular depth estimation. For each model, we visualize the prediction, error to ground-truth, and estimated uncertainty for three randomly chosen examples (A-C). Ideally, the model should predict high uncertainty whenever it does not know the answer (i.e., large error). We evaluate the sensitivity and specificity of the predictive uncertainty in identifying likely errors with ROC curves (D).

ROC curves take into account sensitivity and specificity of the uncertainties towards error predictions and are stronger if they contain greater area under their curve (AUC). Fig. 4D demonstrates that our evidential model provides uncertainty estimates concentrate to where the model is making the errors.

In addition to epistemic uncertainty, we also evaluate the aleatoric uncertainty estimates that are learned from our evidential models as well. Fig. 5 compares the evidential aleatoric uncertainty to those obtained by Gaussian likelihood optimization in several domains with high data uncertainty (mirror reflections and poor illumination). The results between both methods are in strong agreement, identifying mirror reflections and dark regions without visible geometry as sources of high uncertainty.

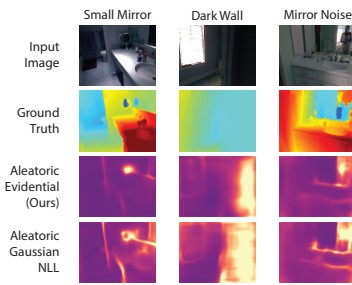

Figure 5: **Aleatoric uncertainty in depth.** Visualizing predicted aleatoric uncertainty in challenging reflection and illumination scenes. Comparison between evidential and (Kendall & Gal, 2017) show strong semantic agreement.

### 4.3 OUT-OF DISTRIBUTION TESTING

A key use of uncertainty estimation is to understand when a model is faced with test samples that fall out-of-distribution (OOD) or when the model's output cannot be trusted. In the previous subsection, we showed that our evidential uncertainties were well calibrated with the model's errors. In this subsection, we investigate the performance on out-of-distribution samples. Fig. 6 illustrates predicted depth on various test input images (left) and outside (right) of the original distribution. All images have not been seen by the model during training. We qualitatively and quantitatively demonstrate that the epistemic uncertainty predicted by our evidential model consistently increases on the OOD samples.

#### 4.3.1 ROBUSTNESS TO ADVERSARIAL SAMPLES

Next, we consider the extreme case of OOD detection where the inputs are adversarially perturbed to inflict maximum error on the model. We compute adversarial perturbations to our test set using the fast gradient sign method (Goodfellow et al., 2014), with increasing scales, $\epsilon$, of noise. Fig. 7A

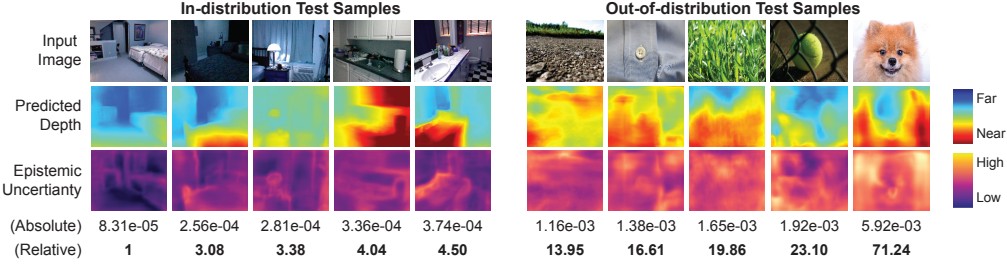

Figure 6: **Out-of-distribution (OOD) data samples.** Evidential models estimate and inflate epistemic uncertainty on OOD data, where the prediction should not be trusted. All samples were not seen during training.

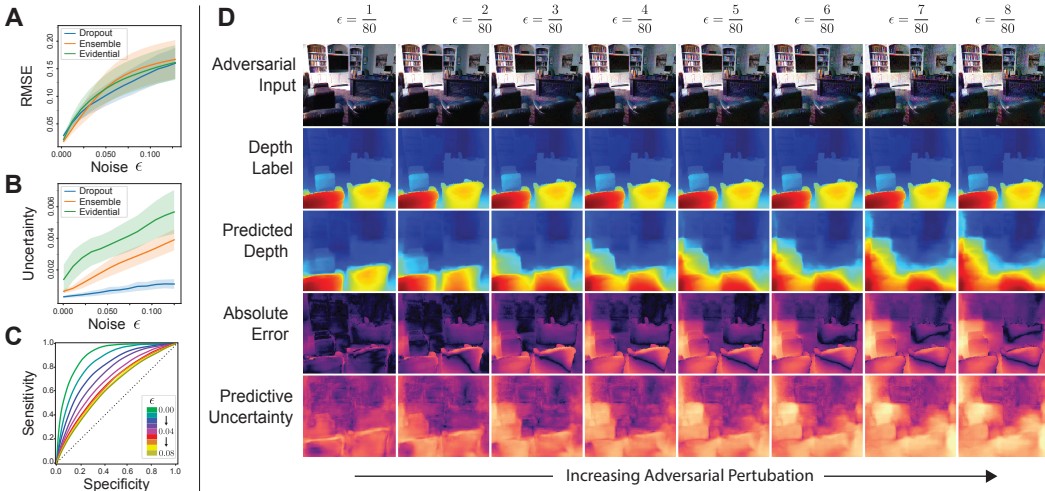

Figure 7: **Evidential robustness under adversarial noise.** Increasing levels of adversarial noise (A) corrupt the predicted depth, and our model begins to incur greater amounts of error. As adversarial noise increases, inferred epistemic uncertainty increases (localized to where the most error occurs). Adversarially perturbed test accuracy (B), epistemic uncertainty (C), as well as the noise to evidential error estimation (D) is also provided.

confirms that the absolute error of all methods increasing as adversarial noise is added. We also observe a positive effect noise on our predictive uncertainty estimates in Fig. 7B. An additional desirable property of evidential uncertainty modeling is that it presents a higher overall uncertainty when presented with adversarial inputs compared to dropout and ensembling methods. Furthermore, we observe this strong overall uncertainty estimation despite the model losing calibration accuracy from the adversarial examples (Fig. 7C).

The robustness of evidential uncertainty against adversarial perturbations is visualized in greater detail in Fig. 7D, which illustrates the predicted depth, error, and estimated pixel-wise uncertainty as we perturb the input image with greater amounts of noise (left-to-right). Note that the predictive uncertainty not only steadily increases as we increase the noise, but the spatial concentrations of uncertainty throughout the image maintain tight correspondence with the error.

## 5   DISCUSSION AND RELATED WORK

Uncertainty estimation has a long history in neural networks, from modeling probability distribution parameters over outputs (Bishop, 1994) to Bayesian deep learning (Kendall & Gal, 2017). Our work builds on this foundation and presents a scalable representation for inferring the parameters of an evidential uncertainty distribution while simultaneously learning regression tasks via MLE.

In Bayesian deep learning, priors are placed over network weights and estimated using variational inference (Kingma et al., 2015). Dropout (Gal & Ghahramani, 2016; Molchanov et al., 2017) and BBB (Blundell et al., 2015) rely on multiple samples to estimate predictive variance. Ensembles (Lakshminarayanan et al., 2017) provide a tangential approach where sampling occurs over multiple trained instances. In contrast, we place uncertainty priors over the likelihood function and thus only need a single forward pass to evaluate both prediction and uncertainty. Additionally, our approach of uncertainty estimation proved to be better calibrated and capable of predicting where the model fails.

A large topic of research in Bayesian inference focuses on placing prior distributions over hierarchical models to estimate uncertainty (Gelman et al., 2006; 2008). Our methodology falls under the class of evidential deep learning which models higher-order distribution priors over neural network predictions to interpret uncertainty. Prior works in this field (Sensoy et al., 2018; Malinin & Gales, 2018) have focused exclusively on modeling uncertainty in the classification domain with Dirichlet prior distributions. Our work extends this field into the broad range of regression learning tasks (e.g. depth estimation, forecasting, robotic control learning, etc.) and demonstrates generalizability to out-of-distribution test samples and complex learning problems.

## 6 CONCLUSION

In this paper, we develop a novel method for training deterministic NNs that both estimates a desired target and evaluates the *evidence* in support of the target to generate robust metrics of model uncertainty. We formalize this in terms of learning evidential distributions, and achieve stable training by penalizing our model for prediction errors that scale with the available evidence. Our approach for evidential regression is validated on a benchmark regression task. We further demonstrate that this method robustly scales to a key task in computer vision, depth estimation, and that the predictive uncertainty increases with increasing out-of-distribution adversarial perturbation. This framework for evidential representation learning provides a means to achieve the precise uncertainty metrics required for robust neural network deployment in safety-critical domains.

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

## 7 APPENDIX

### 7.1 MODEL EVIDENCE DERIVATIONS

For convenience, define $\tau = 1/\sigma^2$ be the precision of a Gaussian distribution. The change of variables transforms the Normal Inverse-Gamma distribution $p(\mu, \sigma^2|\gamma, \lambda, \alpha, \beta)$ to the equivalent Normal Gamma distribution $p(\mu, \tau|\gamma, \lambda, \alpha, \beta)$, parameterized by precision $\tau \in (0, \infty)$ instead of variance $\sigma^2$,

$$p(\mu, \tau|\gamma, \lambda, \alpha, \beta) = \frac{\beta^\alpha \sqrt{\lambda}}{\Gamma(\alpha)\sqrt{2\pi}} \tau^{\alpha - \frac{1}{2}} e^{-\beta\tau} e^{-\frac{\lambda\tau(\mu-\gamma)^2}{2}}. \tag{12}$$

#### 7.1.1 TYPE II MAXIMUM LIKELIHOOD LOSS

Marginalizing out $\mu$ and $\tau$ gives the result of equation 5,

$$p(y_i|m) = \int_\tau \int_\mu p(y_i|\mu, \tau) \, p(\mu, \tau|\gamma, \lambda, \alpha, \beta) \, \mathrm{d}\mu \, \mathrm{d}\tau \tag{13}$$

$$= \int_{\tau=0}^\infty \int_{\mu=-\infty}^\infty \left[ \sqrt{\frac{\tau}{2\pi}} e^{-\frac{\tau}{2}(y_i-\mu)^2} \right] \left[ \frac{\beta^\alpha \sqrt{\lambda}}{\Gamma(\alpha)\sqrt{2\pi}} \tau^{\alpha-\frac{1}{2}} e^{-\beta\tau} e^{-\frac{\lambda\tau(\mu-\gamma)^2}{2}} \right] \mathrm{d}\mu \, \mathrm{d}\tau \tag{14}$$

$$= \int_{\tau=0}^\infty \frac{(\beta\tau)^\alpha}{\Gamma(\alpha)} \sqrt{\frac{\lambda}{2\pi\tau(1+\lambda)}} e^{-\beta\tau} e^{-\frac{\tau\lambda(\gamma-y_i)^2}{2(1-\lambda)}} \, \mathrm{d}\tau \tag{15}$$

$$= 2^{\frac{1}{2}+\alpha} \beta^\alpha \sqrt{\frac{\lambda}{2\pi(1+\lambda)}} \left( 2\beta + \frac{\lambda(\gamma-y_i)^2}{1+\lambda} \right)^{-\frac{1}{2}-\alpha}. \tag{16}$$

For computational reasons it is common to instead minimize the negative logarithm of the model evidence.

$$\mathcal{L}_i^{\mathrm{NLL}}(\boldsymbol{w}) = -\log p(y_i|m) = -\log \left( 2^{\frac{1}{2}+\alpha} \beta^\alpha \sqrt{\frac{\lambda}{2\pi(1+\lambda)}} \left( 2\beta + \frac{\lambda(\gamma-y_i)^2}{1+\lambda} \right)^{-\frac{1}{2}-\alpha} \right) \tag{17}$$

### 7.1.2 SUM OF SQUARES LOSS

Similarly, we can marginalize out $\mu$ and $\sigma^2$ to receive the result of equation 8,

$$\mathcal{L}_i^{\text{SOS}}(\boldsymbol{w}) = \int_{\sigma^2} \int_{\mu} \mathbb{E}_{y \sim p(y|\mu,\sigma^2)} \left[ ||y_i - y||_2^2 \right] p(\mu, \sigma^2 | \gamma, \lambda, \alpha, \beta) \, \mathrm{d}\mu \, \mathrm{d}\sigma^2 \tag{18}$$

$$= \int_{\sigma^2} \int_{\mu} \int_{y} ||y_i - y||_2^2 \, p(y|\mu, \sigma^2) \, p(\mu, \sigma^2 | \gamma, \lambda, \alpha, \beta) \, \mathrm{d}y \, \mathrm{d}\mu \, \mathrm{d}\sigma^2 \tag{19}$$

$$= \int_{\sigma^2=0}^{\infty} \int_{\mu=-\infty}^{\infty} \int_{y=-\infty}^{\infty} ||y_i - y||_2^2 \left[ \sqrt{\frac{1}{2\pi\sigma^2}} e^{-\frac{(y-\mu)^2}{2\sigma^2}} \right]$$

$$\left[ \frac{\beta^\alpha \sqrt{\lambda}}{\Gamma(\alpha)\sqrt{2\pi}} \left( \frac{1}{\sigma^2} \right)^{\alpha+\frac{3}{2}} e^{-\frac{\beta}{\sigma^2}} e^{-\frac{\lambda(\mu-\gamma)^2}{2\sigma^2}} \right] \mathrm{d}y \, \mathrm{d}\mu \, \mathrm{d}\sigma^2 \tag{20}$$

$$= \int_{\sigma^2=0}^{\infty} \int_{\mu=-\infty}^{\infty} \left[ (y_i - \mu)^2 + \sigma^2 \right] \left[ \frac{\beta^\alpha \sqrt{\lambda}}{\Gamma(\alpha)\sqrt{2\pi}} \left( \frac{1}{\sigma^2} \right)^{\alpha+\frac{3}{2}} e^{-\frac{\beta}{\sigma^2}} e^{-\frac{\lambda(\mu-\gamma)^2}{2\sigma^2}} \right] \mathrm{d}\mu \, \mathrm{d}\sigma^2 \tag{21}$$

$$= \int_{\sigma^2=0}^{\infty} \frac{\beta^\alpha}{\lambda\Gamma(\alpha)} \exp\left(-\beta/\sigma^2\right) \sigma^{-2(\alpha+1)} \left(\sigma^2(1+\lambda) + \lambda(y_i - \gamma)^2\right) \mathrm{d}\sigma^2 \tag{22}$$

$$= \left( \frac{\Gamma(\alpha - \frac{1}{2})}{4\,\Gamma(\alpha)\,\lambda\sqrt{\beta}} \right) \left( 2\beta(1+\lambda) + (2\alpha - 1)\lambda(y_i - \gamma)^2 \right) \tag{23}$$

### 7.2 KL-DIVERGENCE OF THE NORMAL INVERSE-GAMMA

The KL-divergence between two Normal Inverse-Gamma functions is given by (Soch & Allefeld, 2016):

$$\mathbb{KL}(p(\mu, \sigma^2 | \gamma_1, \lambda_1, \alpha_1, \beta_1) || p(\mu, \sigma^2 | \gamma_2, \lambda_2, \alpha_2, \beta_2) \tag{24}$$

$$= \frac{1}{2}\frac{\alpha_1}{\beta_1}(\mu_1 - \mu_2)^2 \lambda_2 + \frac{1}{2}\frac{\lambda_2}{\lambda_1} - \frac{1}{2} + \alpha_2 \log\left(\frac{\beta_1}{\beta_2}\right) - \log\left(\frac{\Gamma(\alpha_1)}{\Gamma(\alpha_2)}\right) \tag{25}$$

$$+ (\alpha_1 - \alpha_2)\Psi(\alpha_1) - (\beta_1 - \beta_2)\frac{\alpha_1}{\beta_1} \tag{26}$$

$\Gamma(\cdot)$ is the Gamma function and $\Psi(\cdot)$ is the Digamma function. The evidence is defined by $(2\alpha + \lambda)$. For zero evidence, both $\alpha = 0$ and $\lambda = 0$. To compute the KL divergence between one N.I.G distribution and another with zero evidence we can set either $\{\alpha_2, \lambda_2\} = 0$ (i.e., forward-KL) in which case, $\Gamma(0)$ is not well defined, or $\{\alpha_1, \lambda_1\} = 0$ (i.e. reverse-KL) which causes a divide-by-zero error of $\lambda_1$. In either approach, the KL-divergence between an arbitrary N.I.G and one with zero evidence can not be evaluated.

### 7.3 BENCHMARK REGRESSION TASK EVALUATIONS

#### 7.3.1 EPISTEMIC UNCERTAINTY ESTIMATION

The training set consists of training examples drawn from $y = \sin(3x)/(3x) + \epsilon$, where $\epsilon \sim \mathcal{N}(0, 0.02)$ in the region $-3 \leq x \leq 3$, whereas the test data is unbounded. All models consisted of 100 neurons with 3 hidden layers and were trained to convergence. The data presented in Fig. 3 illustrates the estimated epistemic uncertainty and predicted mean accross the entire test set, $-3 \leq x \leq 3$.

#### 7.3.2 IMPACT OF THE EVIDENTIAL REGULARIZER

In the following experiment, we demonstrate the importance of augmenting the training objective with our evidential regularizer $\mathcal{L}^R$ as introduced in Sec. 3.2. Fig. 8 provides quantitative results on training the same regression problem presented in 7.3.1 with and without this evidential regularization term. This term introduces an "uncertain" prior into our learning process so out-of-distribution (OOD)

samples exhibit high epistemic uncertainty. Without the use of this novel loss term, the learned epistemic uncertainty is unreliable on OOD data.

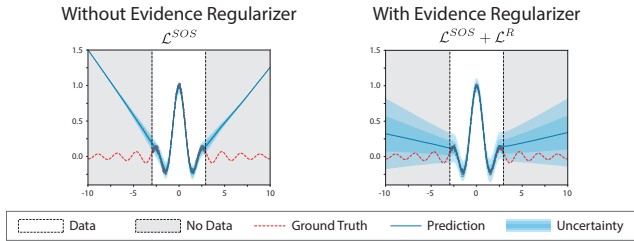

Figure 8: **Evidential regularizer**. The use of our novel $\mathcal{L}^R$ loss during training helps minimize evidence (maximize uncertainty) on out-of-distribution data, thus enabling OOD uncertainty robustness for regression prediction problems.

### 7.3.3 ALEATORIC UNCERTAINTY ESTIMATION

The training set consists of training examples drawn from $y = \sin(3x)/(3x) + \epsilon(x)$, where $\epsilon(x) \sim \mathcal{N}(0, s(x))$, and $s(x) = \frac{1}{20}\cos(3.3x) + 0.1$. We evaluate against (Kendall & Gal, 2017) which presents an algorithm for heteroscedastic aleatoric uncertainty estimation by inferring the mean and variance of a Gaussian likelihood function. As presented in the paper, training is done by minimizing the negative log-likelihood of the data given the inferred likelihood parameters. Both our network and the baseline Gaussian NLL network consisted of 100 neurons with 3 hidden layers and were trained to convergence.

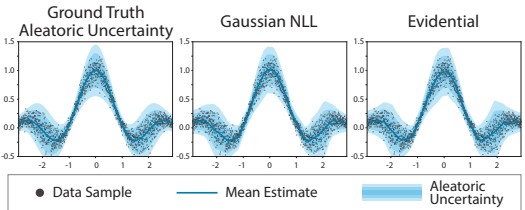

Figure 9: **Aleatoric uncertainty estimation.** Comparing the ability to learn the heteroscedastic aleatoric uncertainty in a synthetic dataset. Evidential modelling is able to match the performance of Gaussian likelihood optimization (Kendall & Gal, 2017).

### 7.4 ADDITIONAL DEPTH ESTIMATION PERFORMANCE RESULTS

| | N | # Parameters | | Inference Speed | | RMSE | NLL |
|---|---|---|---|---|---|---|---|
| | | Absolute | Relative | Seconds | Relative | | |
| **Evidential (Ours)** | - | 7,846,776 | 1.00 | 0.003 | 1.00 | $0.024 \pm 0.032$ | $-1.128 \pm 0.290$ |
| **Spatial Dropout** | 2 | 7,846,657 | 1.00 | 0.028 | 10.20 | $0.033 \pm 0.037$ | $-0.564 \pm 0.231$ |
| **Spatial Dropout** | 5 | 7,846,657 | 1.00 | 0.031 | 11.48 | $0.031 \pm 0.033$ | $-1.227 \pm 0.374$ |
| **Spatial Dropout** | 10 | 7,846,657 | 1.00 | 0.037 | 13.69 | $0.035 \pm 0.042$ | $-1.139 \pm 0.379$ |
| **Spatial Dropout** | 25 | 7,846,657 | 1.00 | 0.065 | 23.99 | $0.032 \pm 0.035$ | $-1.137 \pm 0.327$ |
| **Spatial Dropout** | 50 | 7,846,657 | 1.00 | 0.107 | 39.36 | $0.032 \pm 0.036$ | $-1.110 \pm 0.381$ |
| **Ensembles** | 2 | 15,693,314 | 2.00 | 0.005 | 1.94 | $0.026 \pm 0.032$ | $-1.080 \pm 3.334$ |
| **Ensembles** | 5 | 39,233,285 | 5.00 | 0.010 | 3.72 | $0.023 \pm 0.027$ | $-1.077 \pm 0.298$ |
| **Ensembles** | 10 | 78,466,570 | 10.00 | 0.019 | 6.82 | $0.025 \pm 0.038$ | $-0.980 \pm 0.298$ |
| **Ensembles** | 25 | 196,166,425 | 25.00 | 0.045 | 16.45 | $0.022 \pm 0.029$ | $-1.000 \pm 0.259$ |
| **Ensembles** | 50 | 392,332,850 | 50.00 | 0.112 | 41.26 | $0.022 \pm 0.031$ | $-0.996 \pm 0.275$ |
| **BBB** | 2 | 11,772,929 | 1.50 | 0.064 | 23.58 | - | - |
| **BBB** | 5 | 11,772,929 | 1.50 | 0.070 | 25.55 | - | - |
| **BBB** | 10 | 11,772,929 | 1.50 | 0.088 | 32.25 | - | - |
| **BBB** | 25 | 11,772,929 | 1.50 | 0.144 | 53.09 | - | - |
| **BBB** | 50 | 11,772,929 | 1.50 | 0.284 | 104.33 | - | - |

Table 3: **Depth estimation performance.** Comparison of different epistemic uncertainty estimation algorithms and predictive performance on an unseen test set. Dropout, ensembles, and Bayes-by-Backprop were sampled N times on parallel threads. The evidential method outperforms all other algorithms in terms of space (#Parameters) and inference speed while maintaining competetive RMSE and NLL.

