# OpenReview forum: "Deep Evidential Uncertainty"
_ICLR.cc/2020/Conference — Reject_

### Official Review · AnonReviewer1 · 2019-10-22
**Official Blind Review #1**

**Rating:** 6

**Review:**

This paper proposes a novel approach to estimate the confidence of predictions in a regression setting. The approach starts from the standard modelling assuming iid samples from a Gaussian distribution with unknown mean and variances and places evidential priors (relying on the Dempster-Shafer Theory of Evidence [1] /subjective logic [2]) on those quantities to model uncertainty in a deterministic fashion, i.e. without relying on sampling as most previous approaches. This opens the door to online applications with fully integrated uncertainty estimates.
This is a very relevant topic in deep learning, as deep learning methods are increasingly deployed in safety-critical domains, and I think that this works deserves its place at ICLR.

Pros:
1.	Novel approach to regression (a similar work has been published at NeurIPS last year for classification [3]), but the extension of the work to regression is important.
2.	The experimental results show consistent improvement in performance over a wide base of benchmarks, scales to large vision problems and behaves robustly against adversarial examples.
3.	The presentation of the paper is overall nice, and the Figures are very useful to the general comprehension of the article.
Cons:
1.	The theory of evidence, which is not widely known in the ML community, is not clearly introduced.
I think that the authors should consider adding a section similar to Section 3 of Sensoy et al. [3] should be considered. Currently, the only step explaining the evidential approach that I found was in section 3.1, in a very small paragraph (between “the mean of […] to \lambda + 2\alpha.”). I believe that the article would greatly benefit from a more thorough introduction of concepts linked to the theory of evidence.
2.	The authors briefly mention that KL is not well defined between some NIG distributions (p.5) and propose a custom evidence regularizer, but there’s very little insight given on how this connects to/departs from the ELBO approach.

Other comments/questions:
1.	(p.1)  I’m not sure to fully understand what’s meant by higher-order/lower-order distributions, could you clarify?
2.	(p.3) In section 3.1, the term in the total evidence \phi_j is not defined.
3.	(p.3) Could you comment on the implications of assuming that the estimated distribution can be factorized?
4.	(p.4) Could you comment on the difference that there is between NLL_ML and NLL_SOS from a modelling perspective?
5.	(p.4) The ELBO loss (6) is unclearly defined, and not connected to the direct context. I would suggest moving this to the section 3.3, where the prior p(\theta) used in eq. (6) is actually defined.
6.	(p.4) In equation (6), p_m(y|\theta) isn’t defined, and q(\theta|y) is already parameterized on y if I understand that q(\theta)=p(t\heta|y1,…,yN). Making the conditioning explicit in equation (6) might make the connection to the ELBO clearer.
7.	(p.7) I’m not sure to understand how the calibration of the predictive uncertainty can be tested by the ROC curves if both the uncertainty and estimates error are normalized. Could you also define more clearly what you mean by an “error at a given pixel”?
8.	Spelling & typos:
-	(p.4) There are several typos in equation (8), where tau should be replaced with 1/\sigma^2.
-	(p.8) In the last sentence, there is “ntwork” instead of network.
-	(p.9) There is a typo in the name of Jøsang in the references.
-	(p.10) In equation (13), due to the change of variable, there should be a
-(1/\tau^2) added;
-	(p.10) In equation (14), the \exp(-\lambda*\pi*(…)) should be replaced with \exp(-\lambda*\tau*(…)).

[1] Bahador Khaleghi, Alaa Khamis, Fakhreddine O Karray, and Saiedeh N Razavi. Multisensor data fusion: A review of the state-of-the-art. Information fusion, 14(1):28–44, 2013.
[2] Audun Jøsang. Subjective Logic: A formalism for reasoning under uncertainty. Springer Publishing Company, Incorporated, 2018.
[3] Sensoy, Murat, Lance Kaplan, and Melih Kandemir. "Evidential deep learning to quantify classification uncertainty." Advances in Neural Information Processing Systems. 2018.


**Experience Assessment:**

I have read many papers in this area.

**Review Assessment: Checking Correctness Of Derivations And Theory:**

I carefully checked the derivations and theory.

**Review Assessment: Checking Correctness Of Experiments:**

I assessed the sensibility of the experiments.

**Review Assessment: Thoroughness In Paper Reading:**

I read the paper thoroughly.

---

> ### Author Response · Authors · 2019-11-11
> **Response to AnnonReviewer1**
>
> We would like to thank the reviewer for their positive and detailed review of our paper as well as their suggestions on improving the work’s exposition. We have incorporated and built on many of your comments and suggestions in our latest revision and are currently working to incorporate the remainder in our next revision.
>
> Specifically, we have clarified the definitions of evidence and provided additional experiments on how this evidence is used to evaluate aleatoric in addition to epistemic uncertainties. Regarding the KL uncertainty loss term, we are actively working to strengthen the text and describe how our formulation relates to the original ELBO formulation as they both aim to inflate uncertainty of the posterior. The reviewers suggestions on this are extremely helpful and will be included in the next revision. Our uncertainty inflation loss function also presents a key contribution of this work compared to related works in the classification domain. While in classification, there is a ground truth (and known) likelihood distribution for every sample, in regression the underlying likelihood distribution is not known a priori, thereby motivating our loss function.
>
> Other comments:
> 1. We have added additional clarification in the text. A higher order distribution is one which, when sampled from, yields a distribution over the data. For example, sampling from a Normal Inverse-Gamma distribution yields the parameters (mean, variance) of a Gaussian distribution over the data. Another interpretation stems from conjugate priors in Bayesian probability theory. For instance, the Dirichlet distribution is the conjugate prior of the categorical distribution, which is more in line with the interpretation taken in [1].
>
> 2. Thank you, we’ve corrected this.
>
> 3. The assumption that the posterior can be factorized over the model parameters follows directly from the mean-field approximation [2] used in mean-field variational Bayes which factorizes over the the model’s latent variables. In the context of evidential distributions, the latent variables are exactly the parameters defining the lower-order distribution (in this case a Gaussian with mean, \mu, and variance, \sigma^2).
>
> 4. Excellent question. L_NLL approaches the optimization problem from the lens of an empirical Bayes formulation, where the objective is to *maximize* model evidence, p(y|m). On the other hand, L_SOS aims to *minimize* the sum of squared errors between the evidential prior and the data that would be sampled from the associated likelihood function. We have clarified these points in our updated manuscript.
>
> 5 & 6. Thank you for this constructive comment, we are currently working on incorporating your advice as well as the similar advice of Reviewer 3 to improve the exposition of Sec. 3.
>
> 7. We appreciate the reviewer for pointing out this confusion. We are actively working on incorporating an improved explanation and a step by step algorithm on computing these ROC curves in our next revision.
>
> 8. Thank you, we have corrected these typos in the revised version.
>
>
> [1] A. Malinin and M. Gales. Predictive uncertainty estimation via prior networks. NeurIPS 2018.
> [2] G. Parisi. Statistical field theory. Addison-Wesley, 1988.

---

> > ### Comment · AnonReviewer1 · 2019-11-15
> > **Further comments**
> >
> > Thank you for your additional explanations.
> >
> > I would like to follow up on some of my comments, specifically the other comment 4. I'm still confused about what this means from a *modelling* perspective. Is there any significantly different behavior expected from either of the methods? The question lies more on the interpretation side.
> >
> > I had also a question on the added part on OOD examples. You show that the uncertainty increases as you present OOD examples, but I notice that within the testing set, there is also a great variability in uncertainties (e.g., the rightmost in distribution example has 4.5 times more error compared to the left-most one). Did you observe a sharp threshold in uncertainty increase by presenting OOD examples (e.g. nothing below 10^{-3}) or does it vary smoothly as you present more and more challenging/clearly OOD examples?

---

> > > ### Author Response · Authors · 2019-11-15
> > > **SOS vs NLL and uncertainty increases**
> > >
> > >
> > > Thank you for your followup questions!
> > >
> > > From a modeling perspective, the difference between the two losses can be seen by observing Eq. 14 (NLL) and 21 (SOS). Note that on Eq. 21 the inner integral over the data, y, has been evaluated to allow for a more direct comparison to NLL. We would like to draw attention to the fact that these two equations follow the same form of:
> > >
> > > \int_\theta  [ X * p(theta|m) ]  d\theta
> > >
> > > where X is the only difference between the two equations. In the NLL case, X, this is simply p(y|theta), which naturally is the likelihood function of the data. However, in the SOS case, X interestingly becomes:
> > >
> > > (y - mu)^2 + sigma^2
> > >
> > > Which places an L2 penalty on the data (from the mean), while also trying to drive the aleatoric uncertainty (sigma) to zero. Therefore, the SOS X loss term shown here is optimizing directly on the error of the parameters whereas the NLL X term is trying to maximize the likelihood of the data given the distribution. We hope this explicit term helps bring light to the modelling difference between our two losses, please let us know if there is anything else we can clarify on this front.
> > >
> > > We would like to now turn to your second question on our new OOD example experiment. For a general comment reader, the Figure we will be discussing is Fig. 6. You are certainly correct that epistemic uncertainty increases also on the in-distribution samples we show. In fact, this was exactly what we were trying to demonstrate by sorting the images by uncertainty from left-to-right. We recognize that even when in-distribution there are certain samples which are clearly harder than others and can lead to a greater chance of error. The same can be said for our out-of-distribution examples (i.e. the dog is far more out-of-distribution than the rubble ground). We focus on in-distribution test samples in this response as the reviewer’s question is directly asking about those. Looking at the columns from left-to-right in this figure we can observe that:
> > >
> > > - column 1: is relatively straightforward, a large part of the image is a well-illuminated ground plane, the corners and edges of nearly all objects are clearly shown
> > > - column 2: more challenging than column 1, very poor illumination which can certainly cause an increase in epistemic uncertainty;
> > > - column 3: might seem simple at first, however there is a large unknown object placed on the far right edge of the image that is very difficult to gain a depth understanding of because on a small portion is shown. The lamp is also fairly infrequently seen in our dataset causing some added uncertainty;
> > > - column 4: presents many small objects scattered on top of the kitchen counter, each of these objects is very hard to accurately see and infer depth from;
> > > - column 5: the most challenging in-distribution example that we see. There are many challenges we see with this example, the most obvious being the giant mirror that takes up a significant part of the image. Mirrors are very challenging from both the epistemic and aleatoric point of view (as we show in Fig. 5). Like previous columns there are also many smaller objects that could be attributed to the increase in epistemic uncertainty.
> > >
> > > To directly answer your question, yes we did observe that the uncertainty of in-distribution test images varied; however, we would characterize the changes as generally smooth with very heavy tails which we do not visualize here (in general there were no large jumps outside of the tails). The jump between the first and second column is potentially entering the lower tail region, but we felt it was important to visualize for the reader at least one example where the uncertainty is lower than the remainder of the in-distribution images and with clear semantic reason (well-lit area, image of standard objects, etc).  On the other hand, the change between in-distribution and OOD was not nearly as smooth. There was a clear jump between the most certain OOD example and the most uncertain in-distribution example like the reviewer mentions. The gravel example presents many similarities to our dataset as it contains a main ground plane and depth increases into the horizon; however, we see there is still a large jump between this example and our in-distribution samples.

---

### Official Review · AnonReviewer3 · 2019-10-22
**Official Blind Review #3**

**Rating:** 6

**Review:**

This paper proposed deep evidential regression, a method for training neural networks to not only estimate the output but also the associated evidence in support of that output. The main idea follows the evidential deep learning work proposed in (Sensoy et al., 2018) extending it from the classification regime to the regression regime, by placing evidential priors over the Gaussian likelihood function and performing the type-II maximum likelihood estimation similar to the empirical Bayes method [1,2]. The authors demonstrated that the both the epistemic and aleatoric uncertainties could be estimated in one forward pass under the proposed framework without resorting to multiple passes and showed favorable uncertainty comparing to existing methods. Robustness against out of distribution and adversarially perturbed data is illustrated as well.

On the technical side, the novelty is incremental. The extension from the classification regime to the regression regime, from the conjugate Dirichet prior to the conjugate Normal-Inverse-Gamma prior, is quite straightforward. Besides, the presentation of the paper could be largely improved. It is not easy to follow the derivation in Section 3. The discussion of concepts and problem definitions look fragmented and incoherent. Even though the presentation largely follows (Sensoy et al., 2018) and uses terms from theory of evidence, the derivation actually is more aligned with the prior network [3] under the Bayesian framework which is missing from the references. It is really confusing that the authors talked about the variational inference when conjugate prior is used, and it is unclear how the variational distributions are used in Section 3.2 or how the "I don't know" loss term relates to the KL-divergence between the variational distribution and the prior in Section 3.3. This term was manually added as additional regularization to "prefer the evidence to shrink to zero for a sample if it cannot be correctly classified" in (Sensoy et al., 2018), and a different regularization was used to encourage distributional uncertainty in [3]. I hope that the authors could spend more efforts clarifying their ideas, especially the derivations in Section 3.2 and 3.3.

On the other hand, there is no referring to the input x in the entire derivation and problem formulation in Section 3. It took me a while to realize that the formulation in (4) actually defines the generation for a particular input, not for all the inputs. That is, the model is trying to model heteroscedastic uncertainty, not the homoscedastic counterpart. It could be better to call out the dependence on the input explicitly.

On the quantitative side, the baseline models considered in Section 4 are mainly concerned with epistemic uncertainty estimation. So it would be good to explicitly discuss which uncertainty estimation was compared with. This work estimates both aleatoric and epistemic uncertainties, so a better comparison is to models that estimate both quantities (Kendall & Gal, 2017)[4] which has been shown to give better output estimation comparing to epistemic uncertainty estimation only (Kendall & Gal, 2017).

Other comments:
- What is the \pi in equation (8)?
- The "I don't know" loss introduced in Section 3.3 used L-p norm. What is the originality of the L-p norm here? In practice, which p value should be used? In the experiments, which p value was used?
- The RMSE results of the depth estimation presented in Table 2 are orders of magnitude smaller than those from existing work, for example Table 2(b) in (Kendall & Gal, 2017). Was a different RMSE computation used in this work?
- From the caption in Table 2, it seems that only 5 samples were used in MC-dropout, which is considerably smaller than those used in existing work (Kendall & Gal, 2017).

[1] D.J.C. MacKay. Hyperparameters: optimize, or integrate out? Maximum Entropy and Bayesian Methods, Springer 1996.
[2] B. Efron. Two modeling strategies for empirical Bayes estimation. Statistical Science, 2014.
[3] A. Malinin and M. Gales. Predictive uncertainty estimation via prior networks. NeurIPS 2018.
[4] Y. Kwon, J.-H. Won, B.J. Kim, and M.C. Paik. Uncertainty quantification using Bayesian neural networks in classification: application to ischemic stroke lesion segmentation. MIDL 2018.

**Experience Assessment:**

I have read many papers in this area.

**Review Assessment: Checking Correctness Of Derivations And Theory:**

I carefully checked the derivations and theory.

**Review Assessment: Checking Correctness Of Experiments:**

I carefully checked the experiments.

**Review Assessment: Thoroughness In Paper Reading:**

I read the paper thoroughly.

---

> ### Author Response · Authors · 2019-11-11
> **Response to AnonReviewer 3**
>
> We are grateful for the very detailed and thorough review of our paper, and thank the reviewer for their constructive feedback and prior work references. We would like to clarify several points on novelty and our contribution. While prior works in the classification domain require a ground truth likelihood function over the data, our work, to the best of our knowledge, represents the first demonstration of how epistemic uncertainty can be learned without this information. Thus, our approach enables application to the wider range of regression problems (e.g., depth estimation, forecasting, robotic control learning, etc) which directly map data to targets without a known likelihood function.
>
> Specifically, while the use of a N.I.G. prior is an expected extension from the Dirichlet prior used in the classification regime, there are several challenges in inflating the model’s prior epistemic uncertainty that are specific to regression learning problems. Namely:
>
> 1. To effectively model epistemic uncertainty a regularization loss is needed to minimize divergence to an “uncertain” distribution. In classification [1, 2], this is somewhat trivial and is done by minimizing the KL-divergence from the inferred posterior to a uniform Dirichlet. In the regression domain, a uniform prior is not well defined. A simple univariate example to demonstrate this is that the KL-divergence between any inferred Gaussian and a Gaussian with infinitely large variance is always infinite, regardless of the inferred Gaussian. Therefore, simply inflating uncertainty by minimizing a direct KL loss will not achieve the desired results in regression.
>
> 2. Furthermore both [1, 2] require the inferred distribution to be redistributed and remove the non-misleading evidence. This requires a priori knowledge of the ground truth likelihood function of the data. In classification this is straightforward as the data labels are typically provided as one-hot encodings which directly represents the likelihood function. In regression problems, there unfortunately is no analog as labels directly represent the target with no likelihood association. Our work contributes a solution to handle such regression problems despite classical evidential optimization techniques not being applicable. Furthermore, we acknowledge that inflating the regression uncertainties still presents many open research questions, in addition to the solution presented in this paper.
>
> Therefore, the fundamental learning approaches from the classification domain [1, 2], even after adapting to the context of a N.I.G. prior, are not applicable to regression, thus necessitating the approach presented in this work. We are actively working to improve the exposition of these ideas as well as the introduction and motivation of the “I don’t know” loss term in our next revision of Sec. 3.
>
> We would like to especially thank the reviewer for comments on lack of comparisons to aleatoric uncertainty estimation (in addition to epistemic). As a result, we have conducted numerous additional experiments on synthetic datasets (with known aleatoric uncertainty) as well as on depth dataset. We present our results for both of these experiments and compared to other methods (i.e. [3]) in Figures 5 and 8.
>
> Other comments:
> 1. \pi = 3.1415. \Gamma(.) = Gamma function [4].
>
> 2. We appreciate the reviewer for pointing out this point of confusion in our formulation. We have clarified the choice of p in the text (please refer to pg. 5)
>
> 3. Thank you for raising this point. The output label in our context was the inverse depth (i.e. scaled disparity) which caused the relative differences in error. It is often preferred [5, 6, 7], from an optimization point of view, to predict disparity as opposed to depth as it is more numerically stable (far objects, such as the sky, have an extremely large depth but disparity of zero).
>
> 4. We use a higher number of samples in the accuracy experiments presented in Table 1. On the other hand, Table 2 aimed to compare compute efficiency (not accuracy), so to give sampling techniques an added benefit, we report runtime and memory requirements on a smaller number of samples. We have also added experiments (Table 3) with greater number of samples to address the reviewer’s suggestion.
>
> [1] M. Sensoy, et al. "Evidential deep learning to quantify classification uncertainty." NeurIPS. 2018.
> [2] A. Malinin, et al. Predictive uncertainty estimation via prior networks. NeurIPS 2018.
> [3] A. Kendall, et al. "What uncertainties do we need in bayesian deep learning for computer vision?." NeurIPS. 2017.
> [4] Gamma Function. Wolfram Alpha. http://mathworld.wolfram.com/GammaFunction.html
> [5] C. Godard, et al. "Unsupervised monocular depth estimation with left-right consistency." CVPR. 2017.
> [6] A. Kendall, et al. "Multi-task learning using uncertainty to weigh losses for scene geometry and semantics." CVPR. 2018.
> [7] R. Atienza. "Fast Disparity Estimation using Dense Networks." ICRA. 2018.

---

> > ### Comment · AnonReviewer3 · 2019-11-14
> > **Further comments**
> >
> > Thanks for the detailed response and the additional quantitative results on aleatoric uncertainty estimation.
> >
> > But I do not quite follow the part when the authors argued that "In regression problems, there unfortunately is no analog as labels directly represent the target with no likelihood association. Our work contributes a solution to handle such regression problems despite classical evidential optimization techniques not being applicable." Does this refer to the Gaussian likelihood of the regression target used in the manuscript? Why "no likelihood association"?  The proposed work follows the Prior Network framework naturally with a different instantiation in the regression regime. Also, still not quite follow how the variational Bayesian comes into play in Section 3.1 and 3.2 in the revised version.

---

> > > ### Author Response · Authors · 2019-11-15
> > > **Clarification followup**
> > >
> > > Thank you so much for following up with this question! We completely agree with your assessment that our work follows the Prior Network framework naturally and have updated Sec. 3 in our latest revision to take this into account. Please also refer to our latest comment to all reviewers for a summary of these changes.
> > >
> > > Regarding the unavailability of ground truth likelihood data in regression problems, we hope to address your question here (as well as in the now revised Sec. 3). If you refer to the related work on evidential priors for classification [1, 2] the uncertainty is inflated by minimizing the KL between a modified version of the inferred distribution and an uncertainty prior distribution. However, the point we would like to draw attention to is how the inferred distribution is modified. In both approaches, the authors require the ground truth likelihood of the target to redistribute the density of the posterior before minimizing the KL. This way, evidence is reduced only on the classes which the sample was not part of. In classification, the ground truth labels are samples of the categorical likelihood, so we know the full and bounded set of classes that should lower their evidence.
> > >
> > > [side note] In [1] this requirement is described on page 6, and the modified distribution parameters are denoted as \tilde{\alpha}_i. In [2], this requirement is described on page 6, Eq. 12 and 13.
> > >
> > > In the regression setting, this optimization algorithm would require knowledge of the ground truth likelihood (\mu, \sigma) from which the datapoint, y, was sampled. This is because we cannot reduce evidence everywhere except our single point estimate as this space is infinite and unbounded. One alternative could be to leverage the estimated likelihood (which we can obtain from the NIG) to use instead. However, we found that doing so does not work in practice (validated in both our regression setting as well as the prior works classification setting).
> > >
> > > Therefore, to overcome this limitation, we believe one of our main contributions was a novel loss function for expressing uncertainty on mistakes that can be applied to the regression setting. We hope that this explanation as well as the added revision to Sec 3 (specifically 3.2.2) helps clarify our the novelty of our optimization method.
> > >
> > > [1] M. Sensoy, et al. "Evidential deep learning to quantify classification uncertainty." NeurIPS. 2018.
> > > [2] A. Malinin, et al. Predictive uncertainty estimation via prior networks. NeurIPS 2018.

---

> > > > ### Comment · AnonReviewer3 · 2019-11-15
> > > > **Further comments**
> > > >
> > > > Thanks for the additional explanations. Section 3 in the current version is much cleaner. The authors might want to highlight the loss part in the contributions summary in Section 1 and supporting quantitative results showcasing the importance of including this evidence regularizer.

---

> > > > > ### Author Response · Authors · 2019-11-15
> > > > > **Highlighting the evidential regularizer**
> > > > >
> > > > >
> > > > > Thank you very much for recognizing this improvement of Section 3!
> > > > >
> > > > > Following your suggestion, we have just uploaded another revision which integrates our evidence regularizer as an explicit contribution in Section 1, as well as a new quantitative ablation experiment (Sec 7.3.2) to demonstrate the need to use this loss during training.
> > > > >
> > > > > Please let us know if you have any remaining comments which you would like to see addressed in a revised version.

---

> > > > > > ### Comment · AnonReviewer3 · 2019-11-15
> > > > > > **Thanks for the response**
> > > > > >
> > > > > > Thanks for the new quantitative results. I do not have further questions.

---

### Official Review · AnonReviewer2 · 2019-10-26
**Official Blind Review #2**

**Rating:** 3

**Review:**

This paper investigates the aleatoric uncertainty and epistemic uncertainty in machine learning. The evaluation was performed on benchmark regression tasks. The comparison with other state-of-the-art methods was provided. The evaluation of the robustness against out of distribution and adversarially perturbed test data was performed.

Strength:
1. Experiments were complete. Analyses were provided with useful information.
2. A model with smaller number of parameters was proposed.
3. Computation efficiency was improved.

Weakness:
1. Total evidence and model evidence were defined. The derivation of these evidences should be clarified.
2. Theoretical justification for related methods could be improved.

**Experience Assessment:**

I have published one or two papers in this area.

**Review Assessment: Checking Correctness Of Derivations And Theory:**

I assessed the sensibility of the derivations and theory.

**Review Assessment: Checking Correctness Of Experiments:**

I assessed the sensibility of the experiments.

**Review Assessment: Thoroughness In Paper Reading:**

I made a quick assessment of this paper.

---

> ### Author Response · Authors · 2019-11-11
> **Response to AnonReviewer 2**
>
> We would like to thank the reviewer for their positive feedback and constructive comments. Your comments and suggestions have helped us improve the exposition of our work in the latest revised submission. Additionally, we would like to address the weaknesses noted by the reviewer:
>
> 1. We clarify that “model evidence” [1] (also known as marginalized likelihood) is a term from Bayesian inference that describes the distribution of the observed data marginalized over the model parameters. On the other hand, “total evidence” arises directly from the learned conjugate prior (evidential) distribution [2]. We have clarified both of these points in the text and provided a reference for virtual observations of conjugate prior distributions [2] in support of our total evidence definition.
>
> 2. We appreciate the reviewer’s suggestion and are working to address this point in the text in our next revision as it closely relates to similar suggestions made by Reviewers 1 and 3.
>
>
> [1] D. MacKay. "Bayesian model comparison and backprop nets." Advances in neural information processing systems. 1992.
> [2] M. Jordan. "The exponential family: Conjugate priors." (2009).

---

### Public Comment · ~Pranav_Poduval1 · 2019-09-26
**Beautiful work, would be great if Authors could clarify some doubts**

Probably because I don't quite understand Dempster-Shafer theory, I don't get why same work can't be placed under Bayesian Framework( last I heard there were papers claiming Dempster-Shafer theory was not a generalization of Bayesian Theory). On the other hand- Noise Contrastive Priors for Functional Uncertainty- https://arxiv.org/abs/1807.09289, seemed to be able to do Variational Inference on the output space( except theirs is a Gaussian prior ).

Most people don't understand Dempster-Shafer theory, it would be of great help to all if you could highlight the key difference between Dempster-Shafer theory and Bayesian Theory, and why it is of importance.
Bayesian and Frequentist approaches have become de-facto in DL community for quantifying uncertainty, it would be great if you could make an easy to read papers for newbies. I believe it would also help make this paper far more impactful this way.

Really liked your work, though. Thanks

---

### Public Comment · ~Andrey_Malinin1 · 2019-10-02
**Some related work**

Hello!

Turns out you've submitted something which is incredibly similar to a concept I developed in my PhD Thesis, called Regression Prior Networks. Would be great if you guys cite it :)

Link to thesis:  https://mi.eng.cam.ac.uk/~ mjfg/thesis_am969.pdf (Remove space after ~ )

Overall, your work seems quite nice, I like how you visually represented the Normal-inverse-Gamma distribution. Spent a while thinking how to do that. I'm curious how well the proposed approach generalizes to OOD detection without seeing any OOD training data though.

Cheers,
Andrey Malinin

---

> ### Author Response · Authors · 2019-11-12
> **Out-of-distribution test samples**
>
> Thank you for your positive feedback! We’ve included a reference to your NeurIPS paper in our latest revision as relevant prior art.
>
> To answer your question regarding OOD data, we present experiments evaluating epistemic uncertainty on unseen in-distribution and OOD data (Fig. 6) as well as on the extreme case of adversarially perturbed inputs (Fig. 7) — all of which were not part of the training distribution either.

---

### Author Response · Authors · 2019-11-15
**Summary and general response**

We would like to thank all reviewers for their very thoughtful feedback on our work. We have uploaded a final revision which, we believe, incorporates all of your suggestions and clarifies the questions presented. In summary, our work presents the following contributions:

1. A method for learning evidential distributions by placing priors on our likelihood to model the uncertainty of a regression network (applicable to a wide range of problems in depth estimation, forecasting, or robotic control);
2. A novel loss function for inflating uncertainty when mistakes are made during training, thus enforcing an uncertain prior during the learning process;
3. Demonstration that our model is scalable to extremely high dimensional output spaces (i.e. images) and robust to detecting out-of-distribution data as well as adversarially perturbed examples; and
4. Performance analysis against existing epistemic and aleatoric estimation baselines, demonstrating that our method achieves comparable or better predictive performance while being faster (no sampling) and more memory compact (no ensembles, or weight parameterization).


Through our revisions we would like to summarize a key list of changes which we have incorporated (relevant reviewer prepended for each point):

- [R1,R3] Stronger connection from the Prior Network formulation and a description of the challenges that regression problems present over classification, thus motivating our novel loss function (evidence regularizer).
- [R2] A careful introduction of the evidential terms which we defined.  We have added introductions to the terms “total evidence” and “model evidence” and provide references to support their use.
- [R3] Analysis of the predicted aleatoric uncertainty in addition to the epistemic uncertainty (Fig. 5 & 8).
- [R3] New experiments with greater number of samples of baselines to demonstrate the superior predictive capacity and performance benefits of our approach (Tab. 3, augmenting Tab. 2)
- [R1] New experiments on out-of-distribution samples (Fig. 6), demonstrating applicability to detect OOD samples besides those of which have been adversarially perturbed.
- [R3] Ablation experiment on the evidential regularizer loss to demonstrate the importance of this contribution (Sec. 7.3.2).

---

### Decision · Program_Chairs · 2019-12-19

**Decision:**

Reject

**Comment:**

This paper presents a method for providing uncertainty for deep learning regressors through assigning a notion of evidence to the predictions.  This is done by putting priors on the parameters of the Gaussian outputs of the model and estimating these via an empirical Bayes-like optimization.  The reviewers in general found the methodology sensible although incremental in light of Sensoy et al. and Malinin & Gales but found the experiments thorough.  A comment on the paper pointed out that the approach was very similar to something presented in the thesis of Malinin (it seems unfair to expect the authors to have been aware of this, but the thesis should be cited and not just the paper which is a different contribution).  In discussion, one reviewer raised their score from weak reject to weak accept but the highest scoring reviewer explicitly was not willing to champion the paper and raise their score to accept.  Thus the recommendation here is to reject.  Taking the reviewer feedback into account, incorporating the proposed changes and adding more careful treatment of related work would make this a much stronger submission to a future conference.